# Analysis on synergistic cocontraction of extrinsic finger flexors and extensors during flexion movements: A finite element digital human hand model

Ying Lv[1], Qingli Zheng[1], Xiubin Chen[2], Chunsheng Hou[3], Meiwen An👤[1]*

**1** Institute of Biomedical Engineering, College of Biomedical Engineering, Taiyuan University of Technology, Taiyuan, Shanxi, China, **2** Department of Ultrasound, Shanxi Bethune Hospital,Taiyuan, Shanxi, China, **3** Department of Plastic Surgery, Affiliated Hangzhou First People's Hospital, Zhejiang University School of Medicine, Hangzhou, Zhejiang, China

\* meiwen_an@163.com

**Data Availability Statement:** All relevant data are within the paper and its Supporting Information files.

## Abstract

Fine hand movements require the synergistic contraction of intrinsic and extrinsic muscles to achieve them. In this paper, a Finite Element Digital Human Hand Model (FE-DHHM) containing solid tendons and ligaments and driven by the Muscle-Tendon Junction (MTJ) displacements of FDS, FDP and ED measured by ultrasound imaging was developed. The synergistic contraction of these muscles during the finger flexion movements was analyzed by simulating five sets of finger flexion movements. The results showed that the FDS and FDP contracted together to provide power during the flexion movements, while the ED acted as an antagonist. The peak stresses of the FDS, FDP and ED were all at the joints. In the flexion without resistance, the FDS provided the main driving force, and the FDS and FDP alternated in a "plateau" of muscle force. In the flexion with resistance, the muscle forces of FDS, FDP, and ED were all positively correlated with fingertip forces. The FDS still provided the main driving force, but the stress maxima occurred in the FDP at the DIP joint.

## Introduction

The human hand has the most sophisticated anatomy, including 27 bones, numerous muscles, tendons, ligaments, and other anatomical structures [1], which allows for a variety of complex and delicate movements. Many studies have long been conducted on the motor mechanisms of the hand [2,3], hand diseases [4,5], and bionic applications [6–8]. The main methods for studying the motor mechanism of the hand are muscle electrical signals (EMG) and numerical models of the hand. Among them, studies on EMG focus on the activation and inhibition of hand muscles [9–12], while numerical models of the human hand can explain the behavior of hand muscles in motion from a mechanical perspective [13–15].

The use of the Finite Element Digital Human Hand Model (FE-DHHM) to study hand motion mechanisms has a long history, starting from the earliest with Carrigan [16] and

**Funding:** This study is supported by the National Natural Science Foundation of China (No.11372208, No.31870934), including the cost of CT and ultrasound experiments, customization of the force measurement platform and the paper publication dues. The funders had no role in study design, data collection and analysis, decision to publish, or preparation of the manuscript.

**Competing interests:** The authors have declared that no competing interests exist.

Anderson [17], to build 3D wrist models. These models were built from CT images and contained eight carpal bones and ligaments, usually focusing on the stress transfer in the normal or diseased carpal tunnel. Due to the structural incompleteness, such models had limitations for studying the synergistic contraction of the extrinsic finger flexors and extensors of the hand.

The models that can be used for muscle force analysis are FE-DHHMs containing at least fingers and tendons, and there are two main categories, local and global models. Local models mainly focus on the motion mechanisms of a single finger [18–21], such as the biomechanical model of the index finger developed by Brook et al. [18] and the index finger model containing ED network tendons by Hu et al. [21]. Such models contained structures such as three phalanges, joints, and tendons. The tendon was reduced to a one-dimensional linear unit or a network of multiple linear units. The joints were defined by constraint equations. This type of model can better reflect the control mechanism of tendon on a single finger, which is important for the exploration of finger movement mechanism and diagnosis of related diseases. However, due to the large degree of simplification of such models, it is difficult to reflect the real overall motion mechanism of the hand. Therefore, the establishment and use of a holistic model of the hand has become a hot issue for current research by scholars at home and abroad. Chamoret et al. [22] established an FE-DHHM including bone and skin to analyze the contact/impact of the human hand with a deformable rectangular block. Harih et al. [23] established an FE-DHHM including bone, joint and skin, and used the angular displacement of the joint measured by a motion capture system as the driving force, and analyzed the distribution of stress and contact pressure for gripping action. Their work focused on the ergonomic evaluation of handheld products. It is noteworthy that such FE-DHHMs usually reduced joints to simple hinge connections and did not model soft tissues such as muscles and tendons but used the angular displacement of joints as the driving force. This has led to the weak ability of the model to study the synergistic and antagonistic mechanisms of individual muscles during manual movement. Research in this area is important for clinical diagnosis and treatment of muscle and tendon injuries as well as for rehabilitation training.

In this paper, we have achieved accurate loading of Muscle-Tendon Junction (MTJ) displacements of different muscles under the same movement by combining FE-DHHM established by finite element technique and the MTJ displacements measured by ultrasound imaging, which was used to study the synergistic contraction of Flexor Digitorum Superficialis muscle (FDS), Flexor Digitorum Profundus muscle (FDP) and Extensor Digitorum muscle (ED) in flexion movements.

## Materials and methods

To ensure uniformity of model and MTJ displacement data, all CT scans and ultrasound experiments were performed by the author as the volunteer. The volunteer was healthy a 30-year-old male with no hand disease or associated neurological disorders. All experimental protocols and methods were performed in accordance with relevant guidelines and regulations, and were approved by the biological and medical ethics committee of Taiyuan University of technology.

### Geometric model

The geometric model of the human hand was built based on CT scan image files of the volunteer's right hand by the 3D medical image modeling software MIMICS 19.0. The geometric model included 29 bones, such as 14 phalanges, 5 metacarpals, 8 carpal bones and parts of the ulna and radius; 9 muscles and their tendons, such as the FDS, FDP, ED, flexor pollicis brevis,

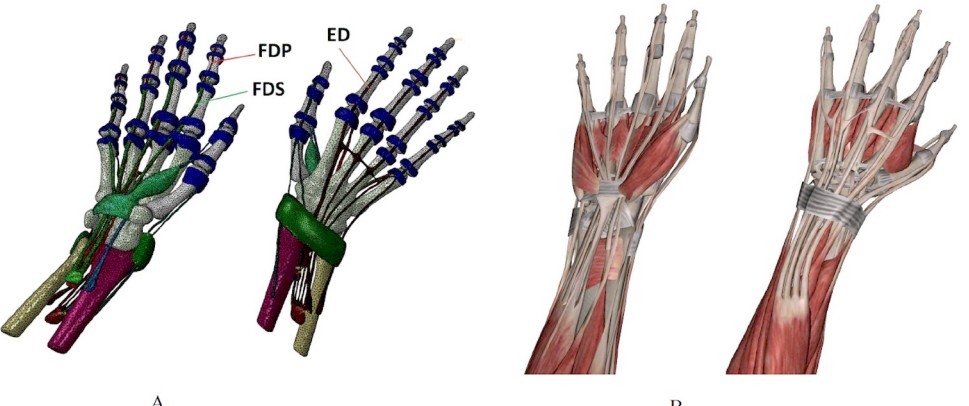

**Fig 1. FE-DHHM and human hand anatomy.** (A) FE-DHHM. (B) human hand anatomy.

flexor pollicis longus, extensor pollicis longus, extensor pollicis brevis, extensor indicis and extensor indicis minimi; ligaments, such as the extensor retinaculum, flexor retinaculum, and annular ligaments that act as finger pulleys at the Interphalangeal (IP) joints and Metacarpophalangeal (MCP) joints (Fig 1).

## Finite element model

The geometric model was smoothed, matched and divided with a tetrahedral mesh C3D4 in 3-matic Medical, generating a total of 375514 elements. The inp files were imported into the finite element software ABAQUS2017 to generate the finite element model with interaction, material parameters, boundary conditions and loading conditions defined.

**(1) Interaction.** The tendons and ligaments were constrained by "Tie" with their corresponding skeletal attachment points. A frictionless self contact was set between each structure. Every joint was connected by three spring elements on the left, right and dorsal sides (simulating the left and right collateral ligaments and dorsal ligaments at the joint) (Fig 2). The three spring elements were all one-dimensional linear elastic elements arranged along the lateral and dorsal midline of the phalanges, which served to maintain joint stability and provide joint stiffness. To simplify the calculation, the following assumptions were made: each spring element at the IP joints had the same spring stiffness, and each spring element at the MCP joints had the same stiffness.

**(2) Material parameters.** To simplify the calculations, the bones, tendons and ligaments were assumed to be linearly elastic isotropic materials (Table 1). In this case, the material parameters of the bones were determined. Due to the discrete nature of soft tissue elastic modulus and errors in model dimensions, the elastic modulus of tendons and ligaments, and the stiffness of spring elements need to be determined from the flexor finger experimental data described below. Therefore, there were four parameters that need to be determined for the model: the elastic modulus of the tendons, the elastic modulus of the ligaments, the IP joint spring elements stiffness, and the MCP joint spring elements stiffness.

**(3) Boundary conditions and loading conditions.** The FDS, FDP and ED in FE-DHHM were split at the MTJ (the location where the cross section increases). The muscles in them were defined as rigid bodies and the MTJ displacement loads and extracted muscle forces were applied at the reference points of the rigid bodies (Fig 2).

The loading conditions were divided into flexion without resistance conditions and flexion with resistance conditions according to the flexion experiment described below. The flexion

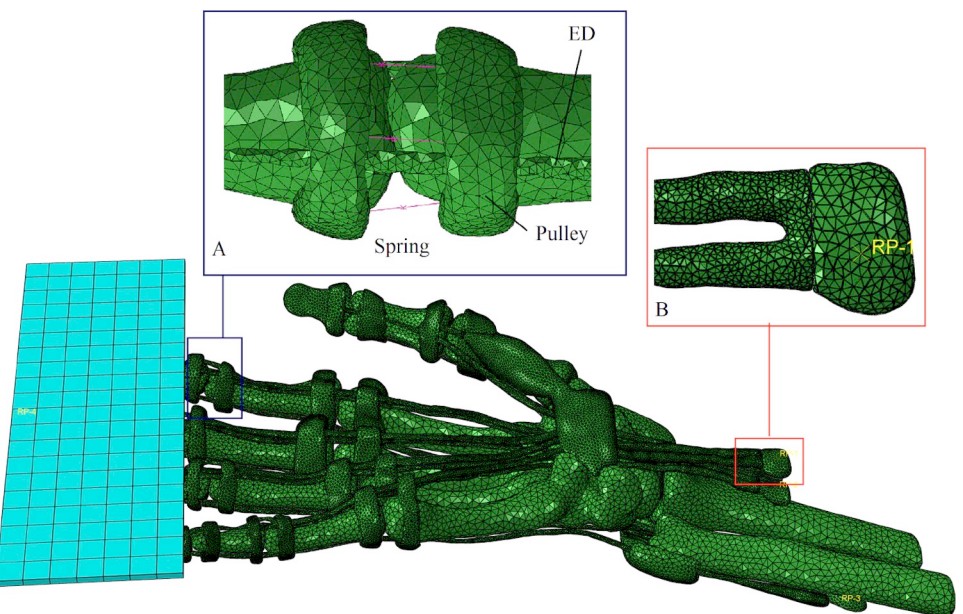

**Fig 2. Loading conditions of flexion with resistance. (A)** The joints were connected by three spring elements. (**B**) The muscle connected to the tendon was defined as a rigid body.

without resistance conditions were based on fixing the metacarpals, carpal bones, ulna and radius of FE-DHHM and loading displacement loads on the rigid reference points of FDS, FDP and ED. The flexion with resistance conditions were based on the flexion without resistance conditions with the proximal phalanges fixed and a fully fixed rigid plate added at the tip of the FE-DHHM to provide resistance (Fig 2). The model was calculated by the dynamic display algorithm.

## Experimental measurement of MTJ displacements in flexion movements

**Grouping of flexion movements.** A force measurement platform was designed including a base plate, a movable steel plate, a fixed steel plate, guide rails and a pressure transducer (Fig 3). When the movable steel plate slid upward along the rail, the average value of the resistance was 0.7N, including the gravity of the movable steel plate and the frictional force between the movable steel plate and the rail. The pressure between the pressure transducer and the movable steel plate was displayed on the monitor when the hand was flat on the platform base plate and bent up to support the movable steel plate. The pressure between the finger and the movable steel plate was called the fingertip force in the text and was equal to the sum of the display and the resistance.

The volunteer was seated with the right arm horizontally on the experimental table, palm up. The palm of the hand in the straightened position was the initial position; and the hand in the naturally relaxed position was the resting position. The finger flexion movements consisted

**Table 1. The preset values of the material parameters.**

| Material | Young's modulus (MPa) | Poisson ratio | Density (Kg/m³) |
|---|---|---|---|
| Bone [24] | 17000 | 0.3 | 2000 |
| Tendon [20] | 125.31 | 0.45 | 1000 |
| Ligament [20] | 114.03 | 0.45 | 1000 |

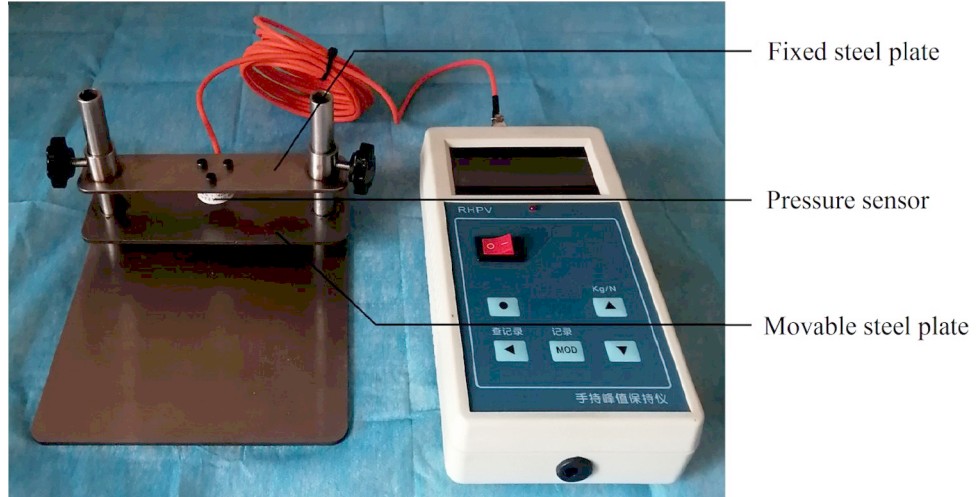

**Fig 3. The force measurement platform.**

of 5 sets of movements, including 1 set of flexion without resistance and 4 sets of flexion with resistance.

Action 1 was flexion without resistance: The hand was flexed from the initial position to the resting position.

Actions 2–5 were flexion with resistance: The hand was placed flat on the base plate of the force measurement platform. The movable steel plate was adjusted to a position just in contact with the finger belly and fixed to the rail with bolts. The hand was flexed up from the initial position to supporting the movable plate until the display showed 5N, 10N, 15N and 20N in sequence, which meant that the fingertip force was 5.7N, 10.7N, 15.7N and 20.7N.

## Measurement of the MTJ displacements by ultrasound

The contraction deformation of the muscle is transmitted to the corresponding bone through the tendon, and the total deformation is reflected in the tendon as the displacement at the MTJ [25,26], which is referred to as the MTJ displacement in the text.The MTJ displacement is divided into two parts: one is the displacement of the tendon due to the change in position between the bones, and the other is the tensile deformation of the tendon during force transmission. When the muscle is actively contracted, the MTJ displacement is the sum of the two; when the muscle is passively stretched, the MTJ displacement is the difference between the two.

Experimental steps: firstly, the ultrasound probe was swept along the forearm longitudinally to locate the target tendon; then the ultrasound probe was swept along the target tendon transversely to locate the location where the cross-section of the tendon becomes larger, which was the MTJ, and the location of the ultrasound probe was marked on the skin; finally, the location of the ultrasound probe before and after the target action was marked and the distance was measured, which was the MTJ displacement of the target action.

We measured the MTJ displacements of the ED, FDS, and FDP in five groups of flexion movements using ultrasound imaging. Fig 4 demonstrates the localization of the FDS tendon and its MTJ. Fig 5 demonstrates the measurement process of the MTJ displacement of the FDS in action 1.

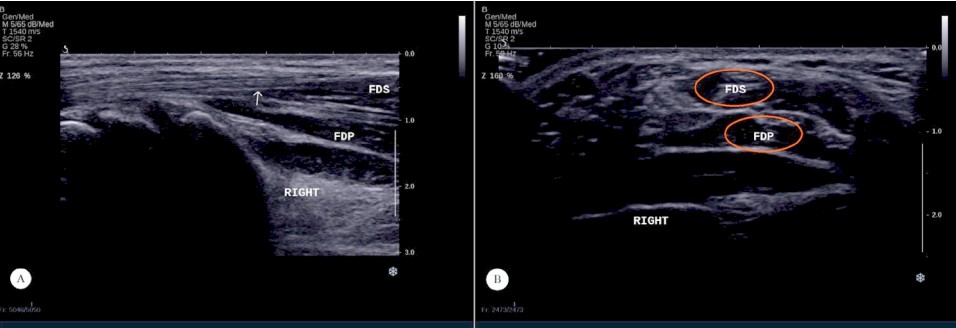

**Fig 4. The positioning of the MTJ for FDS.** (A) Longitudinal section of FDS tendon. (B) Cross sectional section of the MTJ for FDS.

The MTJ displacement of ED in actions 2, 3, 4, and 5 was not significant and was approximated as zero displacement to simplify the calculation. We define the MTJ displacement that decreases the muscle length as positive and the MTJ displacement that increases the muscle length as negative.

## Determination of material parameters and model validation

There were four parameters that need to be determined for the model: the elastic modulus of the tendons, the elastic modulus of the ligaments, the IP joint spring elements stiffness, and the MCP joint spring elements stiffness. The elastic modulus of the tendons directly determined the effect of MTJ displacement and was positively correlated with the fingertip force; the elastic modulus of the ligaments determined the effect of its restraint on the tendons and was also indirectly positively correlated with the fingertip force; the role of the joint spring elements were to maintain joint stability and provide joint stiffness and were negatively correlated with the fingertip force. The known quantities measured experimentally were MTJ displacement and fingertip force (or flexion pattern) in five actions. The four sets of experimental data (MTJ displacement-fingertip force) from actions 2–5 were used to determine the four parameters. The experimental data for action 1 (MTJ displacement-flexion pattern) were used to validate the model after the parameters were determined.

Determination of material parameters: The MTJ displacements of actions 2–5 were input into the model as displacement loads, and after parameter adjustment and feedback calculations, the rigid plate reaction forces (fingertip forces calculated by the model) in FE-DHHM was made equal to the fingertip forces of the force measuring platform in the experiment, so that each parameter satisfying the accuracy was finally determined.

Model validation: The MTJ displacement of action 1 was input into FE-DHHM as displacement load after determining the parameters. The model was validated by comparing the

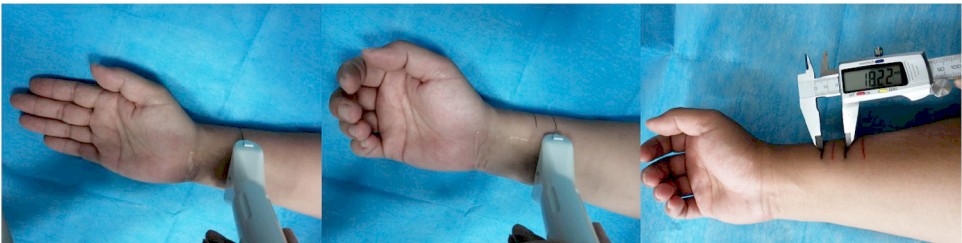

**Fig 5. Measurement procedure of tendon displacement of FDS for action 1.**

flexion pattern of FE-DHHM with that of the hand in the experiment. Table 1 shows the preset values of the material parameters.

# Results

## Determination of material parameters and model validation

Table 2 demonstrates the MTJ displacements of each muscle for five sets of flexion movements, where the ED was elongated in action 1, so the MTJ displacement of the ED was negative. The MTJ displacements from action 2–5 were input to the corresponding FE-DHHM as displacement loads, and the elastic modulus of the tendons and ligaments as well as the spring stiffness were adjusted until the rigid plate reaction forces of the model were equal to the experimental fingertip forces (Table 3).

As shown in Table 3, after adjusting the parameters, the error between the rigid plate reaction forces and the experimental fingertip forces is between 0.48% and 17.8%, which is within the tolerable range. The material parameters thus determined are shown in Tables 4 and 5.

As can be seen in Table 4, the elastic modulus of both tendons and ligaments decreased substantially compared to the preset values. The FE-DHHM with parameters determined was validated with the loading conditions of action 1 (Fig 6).

The flexion process of the FE-DHHM under the loading conditions of action 1 after determining the parameters (Fig 6) is consistent with the experimental flexion without resistance (Fig 5). The model was validated.

## Stress cloud of the tendons

The peak stresses of FDS, FDP, and ED were 23.1 Mpa, 15.6 Mpa, and 12.8 Mpa for the displacement load condition of action 1. The peak stresses of FDS, FDP, and ED were 23.2 Mpa, 32.5 Mpa, and 9.1 Mpa for the displacement load condition of action 5. And the peak stresses of FDS, FDP, and ED in the flexion movements were all at the joints (Fig 7).

The variation of peak stresses with muscle forces for the three muscles is shown in Fig 8. In the flexion without resistance, $F_{FDS} > F_{FDP} > F_{ED}$, and $S_{FDS,Max} > S_{FDP,Max} > S_{ED,Max}$. In the flexion with resistance, $F_{FDS} > F_{FDP} > F_{ED}$, and $S_{FDP,Max} > S_{FDS,Max} > S_{ED,Max}$.

## Muscle force and fingertip force

Fig 9 shows that in the flexion movements, the FDS and FD contracted together to provide power; while the ED had a non-zero muscle force and acted as an antagonist. The FDS produced larger muscle forces with smaller MTJ displacements than the FDP. The muscle forces of FDS, FDP and ED were all positively correlated with fingertip forces during the flexion with resistance. The proportion of muscle force was greatest for FDS and gradually increased with increasing fingertip force, while the proportion of muscle force gradually decreased for FDP and ED. Combined with the MTJ displacements data in Table 2, it was found that both FDS

**Table 2. MTJ displacements of each muscle during flexion movements (mm).**

| Flexion movements | FDS | FDP | ED |
|---|---|---|---|
| Action 1 | 18.22 | 22.22 | -12.54 |
| Action 2 | 4.33 | 6.60 | 0 |
| Action 3 | 10.48 | 12.10 | 0 |
| Action 4 | 15.88 | 16.30 | 0 |
| Action 5 | 20.56 | 21.70 | 0 |

**Table 3. Experimental fingertip forces and the rigid plate reaction forces (N).**

| Flexion movements | Fingertip forces | Reaction forces |
|---|---|---|
| Action 2 | 5.7 | 6.4 |
| Action 3 | 10.7 | 12.6 |
| Action 4 | 15.7 | 13.7 |
| Action 5 | 20.7 | 20.6 |

and FDP produced larger muscle forces with smaller MTJ displacements during flexion with resistance compared to flexion without resistance.

According to Fig 10, it can be seen that in the flexion without resistance (action 1), the muscle forces of FDS and FDP both had a significant plateau period during the increase with time. In contrast, the muscle forces of FDS, FDP and ED were positively correlated with time in the flexion with resistance (actions 2–5), and the fingertip forces were also roughly positively correlated with time under small fluctuations.

## Discussion

The material parameters of the model were determined by the loading conditions of actions 2–5. The determined values of the elastic modulus of both tendons and ligaments were substantially lower than the preset values, but still within the reported range [27]. The reason for the error is that the tendon and ligament models have larger dimensions than the actual ones. Especially for the ligaments, they do not directly determine the joint angle displacements, and their role is to constrain the position of the tendon. Therefore errors in their material parameters have a limited negative impact on the calculation results. Furthermore, it has been shown [28] that: when the point of force application was at the distal phalanx, the extrinsic muscles are the main contributors to joint flexion of the DIP, PIP and MCP joints (accounting for more than 80% of the total force of all flexors); and that the effects of the extensor mechanism on the flexors were relatively small when the location of force application was distal to the PIP joint. The target task (fingertip force) addressed in this paper is consistent with it, so the simplification of the intrinsic muscles and extensor mechanisms in the model of this paper is justified.

Combining Figs 7 and 8, it can be seen that the muscle forces of FDP were smaller than those of FDS in the flexion with resistance, while the peak stresses were larger than those of FDS, and the peak stresses all appeared at the DIP joint of the index finger. In the working condition of resistance flexion in the model, the DIP joint was the interphalangeal joint with the largest angular displacement, and the FDP was the only flexor muscle that crossed the DIP joint. Although the muscle forces in the FDS were larger, the stress distribution was uniform and there was no stress concentration. This indicates that joint flexion has a significant effect on the stress distribution of the flexor tendons.

The torque produced by the muscle-tendon force on the joint is fundamentally influenced by the moment arm (MA), which is defined as the vertical distance between the center of

**Table 4. The determined values of material parameters.**

| Material | Young's modulus (MPa) | Poisson ratio | Density (Kg/m$^3$) |
|---|---|---|---|
| Bone | 17000 | 0.3 | 2000 |
| Tendon | 68 | 0.45 | 1000 |
| Ligament | 20 | 0.45 | 1000 |

**Table 5. Stiffness of the spring unit at the joints.**

| Joints | DIP | | PIP | | MCP | |
|---|---|---|---|---|---|---|
| | Left/Right | posterior | Left/Right | posterior | Left/Right | posterior |
| Stiffness (N/m) | 1000 | 1000 | 1000 | 1000 | 2000 | 2000 |

DIP: Distal interphalangeal joints.

PIP: Proximal interphalangeal joints.

MCP: Metacarpophalangeal joints.

rotation of the joint and the line of action of the muscle-tendon force [29]. One popular technique for estimating MA values is the tendon excursion method, which calculates the instantaneous MA based on the slope of tendon displacement versus joint angle [30,31]. In the present study of the flexion without resistance, the MCP, PIP, and DIP joints were angularly displaced by the combined muscle forces of FDS, FDP, and ED. In the current model's loading conditions, the displacement loads were loaded uniformly. Initially, the MCP joint rotated at the largest angular velocity of the three joints due to the combined forces of the FDS and FDP. This was followed by the PIP joint rotating rapidly due to the rapidly increasing MA of the FDS and FDP on it. Finally, the DIP joint rotated rapidly due to the rapidly increasing MA of FDP on it. The difference in angular velocity of the three joints caused the FDS and FDP in turn to produce a structural MTJ displacement due to the change in the spatial position of the finger. With this displacement, the length of the tendon was constant and therefore the calculated muscle force did not increase. This resulted in a significant plateau in muscle force over time for both FDS and FDP in the flexion without resistance (Fig 10).

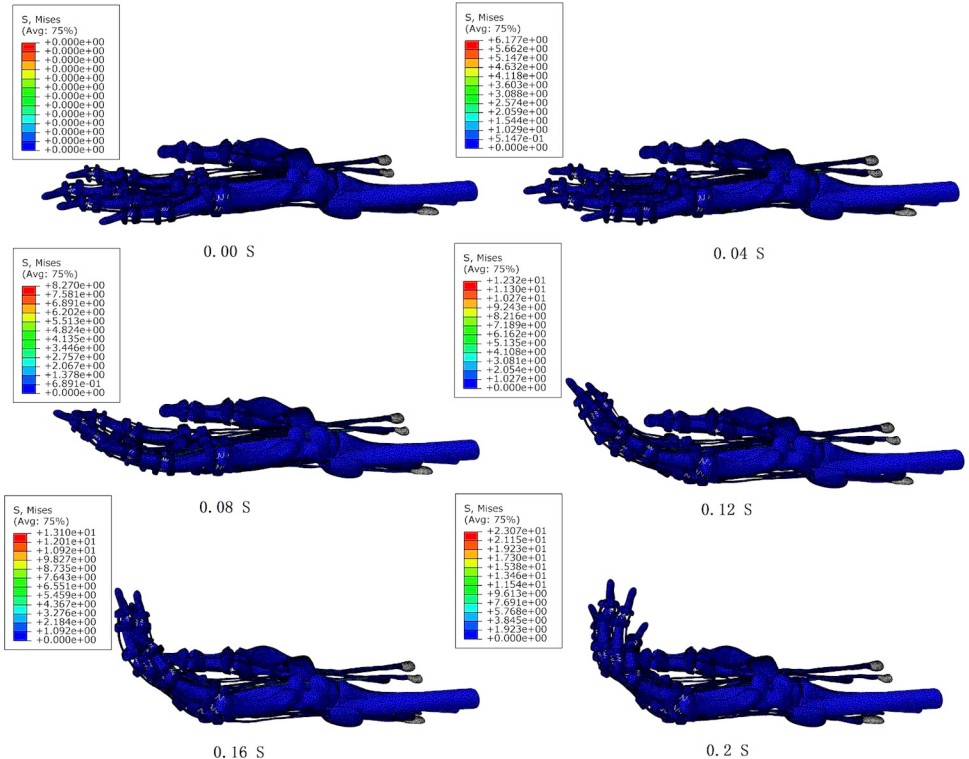

**Fig 6. The process of flexion without resistance.**

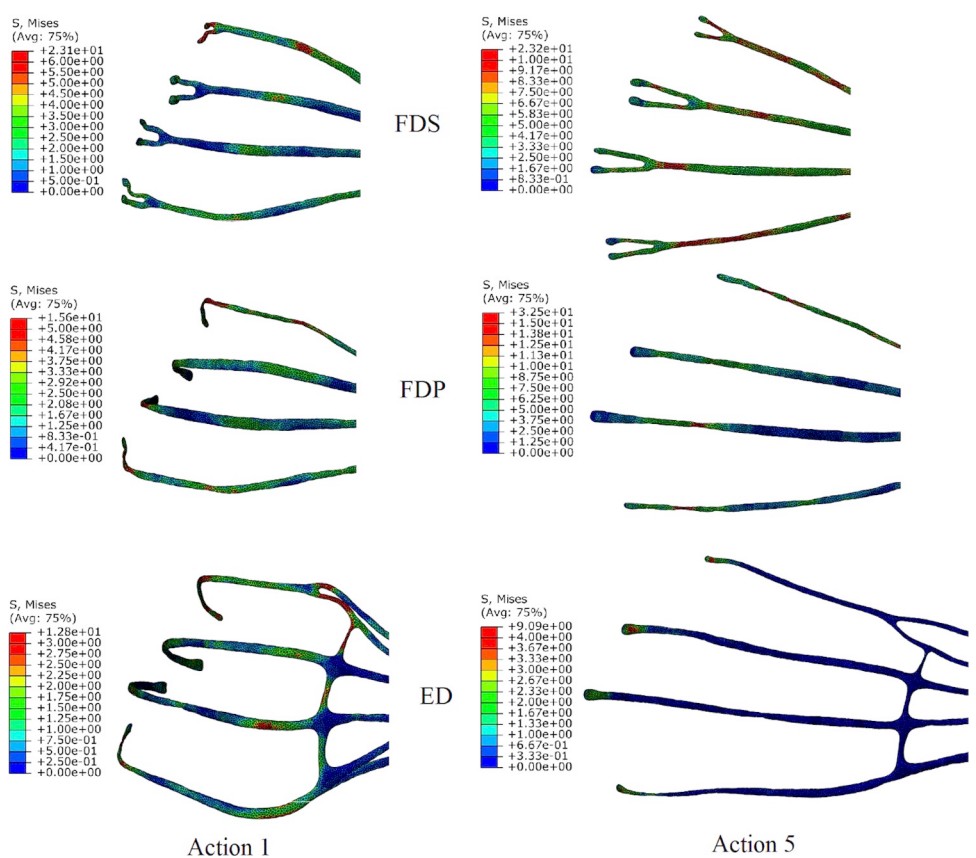

**Fig 7. Stress clouds of each tendon under load conditions of actions 1 and 5.**

In the flexion with resistance, the muscle forces of FDS, FDP and ED were all positively correlated with time. During action 2, the muscle forces of FDS and FDP accounted for 48.9% and 51.1% of the total flexor force, respectively. This value was 62.6% and 37.4% during action 3 and remained stable during action 4 and action 5. Fluctuations in fingertip force over time were caused by changes in the angle of contact between the distal phalanx and the rigid plate in the model.

The ED acted as an antagonist muscle throughout the flexion movements with much smaller muscle forces than the FDS and FDP. The co-activation of the antagonist muscle can improve the precision of the movement and is also important for maintaining joint stability [32]. In a more popular research approach, the role of ED in flexion movements is elaborated as an extensor mechanism (EM) [33,34]. The so-called extensor mechanism is a complex network of tendons connecting the intrinsic and extrinsic muscles of the finger, which increases the maximum fingertip force over a wide range of postures and force directions, allowing for greater finger dexterity during grip. These studies provide new ideas for the refinement of FE-DHHM in this paper.

Schuind et al. [35] measured in vivo the tendon forces generated by the FDS and FDP during passive and active flexion of the index finger in five patients with carpal tunnel syndrome with force transducers. Among them, the active flexion without resistance of the index finger PIP was the result of FDS contraction with some involvement of FDP. The range of FDS tendon force was 3–13 N with a mean of 9 N. The active flexion without resistance of the index finger DIP was due to the contraction of the FDP, which had a muscle force range of 1–29 N

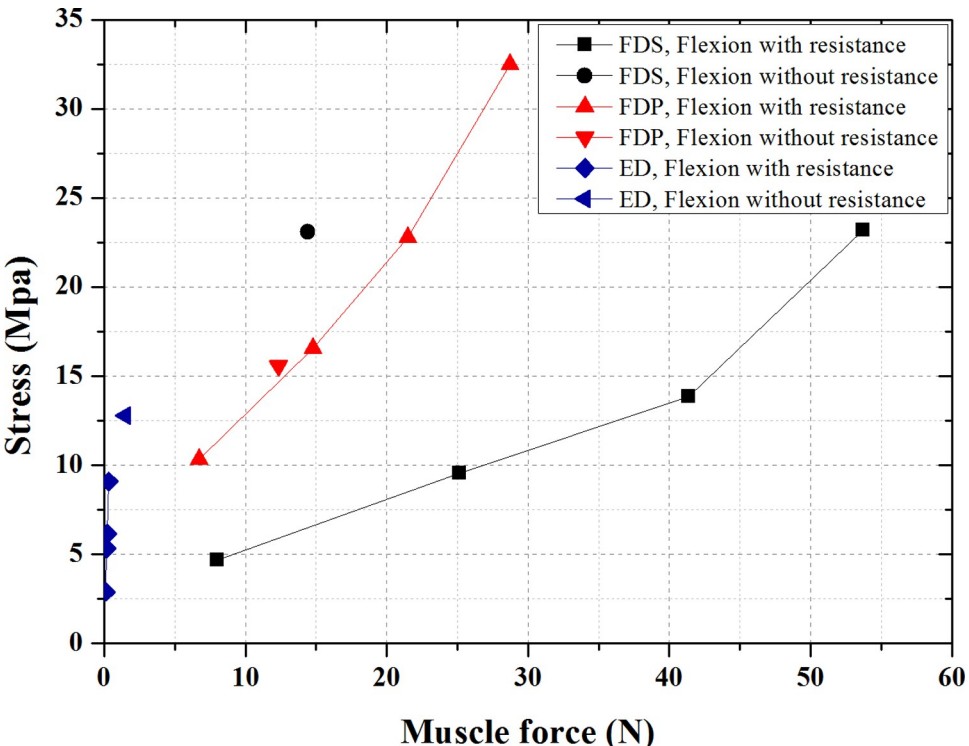

**Fig 8. Variation of peak stresses of three muscles with muscle force during flexion movements.**

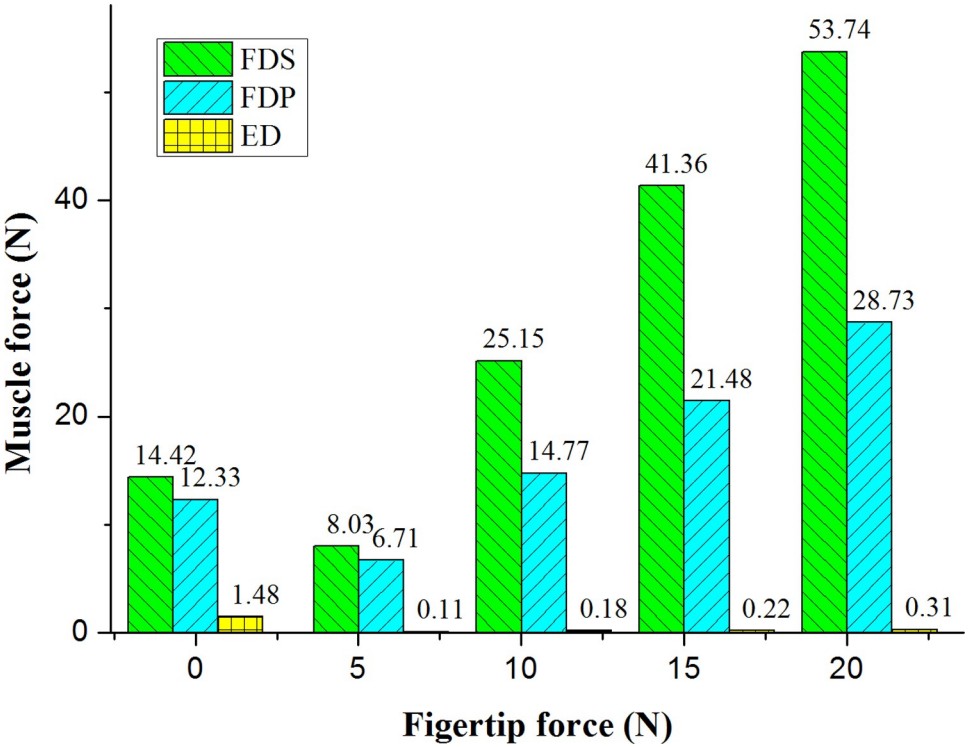

**Fig 9. Variation of muscle forces of the three muscles with fingertip forces during flexion movements.**

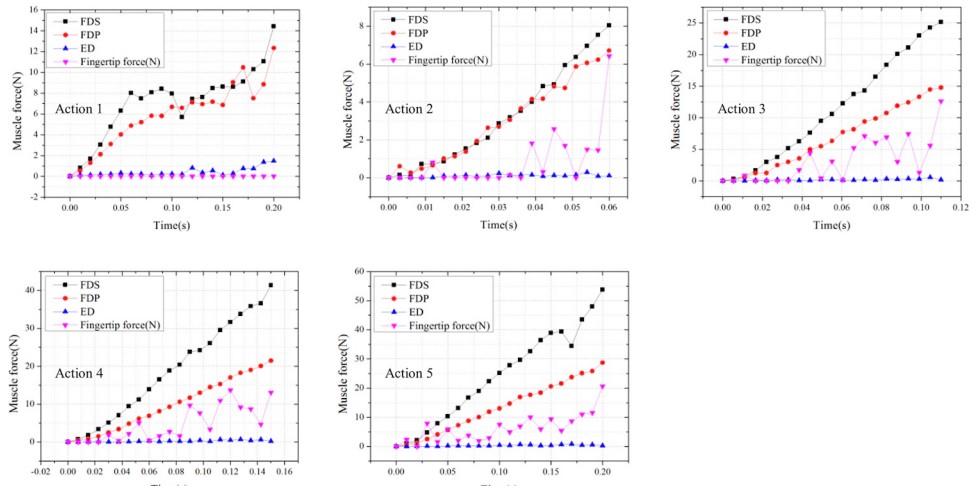

**Fig 10. Variation of muscle forces and fingertip forces over time in three muscles during flexion movements.**

with an average of 19 N. The muscle forces of the FDS and FDP in active flexion without resistance of the four fingers calculated in this paper were 14.42 N and 12.33 N, which were similar to the measurements in the literature.

Kursa et al. [36] measured in vivo the ratio of FDS and FDP tendon force to fingertip force in 15 subjects scheduled for open carpal tunnel surgery when the load cells were pressed with the index finger at different rates up to 15 N. The ratio of FDS tendon force to fingertip force for all tests averaged 1.5 ± 1.0, while the corresponding ratio for FDP averaged 2.4 ± 0.7. In our calculations, the corresponding values were 2.63 and 1.37, which were similar to the measurements in the literature.

The noteworthy difference is that in all the above-mentioned measurements in the literature, the muscle force of the FDS was smaller than that of the FDP, whereas our calculations yielded the opposite result: the FDS produced larger muscle forces with smaller MTJ displacements than the FDP in both the flexion movements with and without resistance. Possible factors contributing to the discrepancy: 1. Mode of movement; the movements studied in this paper were simultaneous flexion of all four fingers, whereas studies in the literatures have targeted the flexion of the index finger alone. This has been verified in the work of Allouch S et al. [37] on the muscle forces during a hand opening-closing paradigm: the muscle forces of the FDS were consistently greater than those of the FDP throughout the movements. 2, Finger posture; it has been noted that finger posture [38–40] and tendon loading conditions [41] could affect fingertip forces. The finger flexion movements in this paper took the palm extension state as the initial position, when both FDS and FDP were passively stretched. In contrast, the flexion movements in the literature all started with the resting position. 3. The intrinsic model of the tendon; to simplify the calculation, the intrinsic model of the tendon with linear elasticity was chosen for the FE-DHHM, which negatively affected the analysis of the tendon with large deformation. How to incorporate the active contraction intrinsic model of the muscle will be the content of our future work.

This study has focused on three extrinsic muscles among the many involved in finger flexion movements. Finger flexion is accomplished by the synergistic contraction of the FDP, FDS, and ED. In order to control external force output and finger position, other muscles must be activated to maintain postural stability and provide proper torque at all joints, including numerous intrinsic and extrinsic muscles. Their respective contributions and roles may

vary depending on the force and finger posture [42]. In addition, clinical practice requires data support in this area, such as the choice of traction force during claw hand traction orthodontics [43]. Therefore, future work requires a more refined FE-DHHM, including more precise construction and more rational material parameters, with the aim of playing a broader role in the field of clinical deformed hand correction or motor rehabilitation.

## Conclusion

The FE-DHHM, which contains solid tendons and ligaments, is a prerequisite for the analysis of individual muscle collaboration and antagonism mechanisms using MTJ displacements as the driving forces. Five sets of MTJ displacements for flexion movements were used to complete the determination of material parameters and validation of validity for the FE-DHHM, and analysis of muscle forces for the external muscles. The model calculations have quantified the contribution of FDS, FDP and ED in flexion movements and elaborated the details of the behavior of each muscle in this process. These phenomena were reasonably explained by comparison with the literatures. The FE-DHHM established in this paper can analyze the synergistic contraction of FDS, FDP and ED, and also has a wide range of roles in medical and rehabilitation fields.

## Author Contributions

**Conceptualization:** Chunsheng Hou, Meiwen An.

**Formal analysis:** Ying Lv.

**Funding acquisition:** Meiwen An.

**Methodology:** Ying Lv, Qingli Zheng, Xiubin Chen.

**Resources:** Chunsheng Hou.

**Software:** Ying Lv.

**Writing – original draft:** Ying Lv.

**Writing – review & editing:** Qingli Zheng.

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
