## [Decision Letter · Decision Letter 0]

23 Oct 2021

PONE-D-21-23164Analysis on synergistic cocontraction of extrinsic finger flexors and extensors during Flexion movements: a Finite Element Digital Human Hand ModelPLOS ONE

Dear Dr. An,

Thank you for submitting your manuscript to PLOS ONE. After careful consideration, we feel that it has merit but does not meet PLOS ONE’s publication criteria as it currently stands. If you feel you can address all the points and concerns raised by the reviewers, some of which are fairly fundamental, a revised version of the manuscript may be considered. 

If you decide to revise, please submit your revised manuscript by Dec 07 2021 11:59PM. If you will need more time than this to complete your revisions, please reply to this message or contact the journal office at plosone@plos.org. Please include the following items when submitting your revised manuscript:A rebuttal letter that responds to each point raised by the academic editor and reviewer(s). You should upload this letter as a separate file labeled 'Response to Reviewers'.A marked-up copy of your manuscript that highlights changes made to the original version. You should upload this as a separate file labeled 'Revised Manuscript with Track Changes'.An unmarked version of your revised paper without tracked changes. You should upload this as a separate file labeled 'Manuscript'.

We look forward to receiving your revised manuscript.

Kind regards,

Xudong Zhang

Academic Editor

PLOS ONE

Journal Requirements:

“The support from the National Natural Science Foundation of China (No.11372208, No.31870934, No.11972243) was acknowledged.”

5. Thank you for your submission to PLOS ONE. Before we can proceed, we kindly ask you to address the following concerns:

We understand that the author served as the volunteer in the study; however, we ask you to present this information in the Methods section of the manuscript. Please revise your Methods section to state all information about ethics committee approval in this section (including the name of the ethics committee), state that the volunteer was the author of the paper, and state any exclusion/inclusion criteria and any relevant demographic information (sex, age, etc.). We appreciate your attention to these queries and look forward to your response.

Reviewers' comments:

Reviewer's Responses to Questions

**Comments to the Author**

1. Is the manuscript technically sound, and do the data support the conclusions?

Reviewer #1: Partly

Reviewer #2: No

2. Has the statistical analysis been performed appropriately and rigorously? 

Reviewer #1: N/A

Reviewer #2: N/A

3. Have the authors made all data underlying the findings in their manuscript fully available?

Reviewer #1: No

Reviewer #2: Yes

4. Is the manuscript presented in an intelligible fashion and written in standard English?

Reviewer #1: Yes

Reviewer #2: No

5. Review Comments to the Author

Reviewer #1: Overview

In this study, the authors created a finite element model of the musculoskeletal structure of the human fingers, which was used to estimate muscle forces during finger flexion. The proposed method incorporates some interesting approach (measuring musculotendon junction excursion), but there are many fundamental problems in the model structure and its validation, which makes it impossible to appreciate its validity or usability. The reviewer thinks that the authors may want to focus on utilizing the information obtained from ultrasound imaging and correlate this to the changes in the measured fingertip forces – FE model used in this study is too complex and contains too many unknown parameters to result in any meaningful estimation of muscle forces.

Major comments

1. Authors adopted an interesting approach, looking at the excursion of the musculotendon junction using ultrasound imaging. This could provide important information regarding the action of different muscles during movements. Unfortunately, there are numerous problems with the modeling approach, and the model validation was performed properly, which makes it very difficult to test the validity of the proposed model. For instance, intrinsic hand muscles are not considered (which are critical in force production), and many model parameters are not properly determined. Some important anatomical features, such as the extensor mechanism or finger pulleys, are not even included in the model.

2. More importantly, the model validation seems to be fundamentally flawed. The only measurements made in this study were musculotendon junction (MTJ) movements and fingertip forces (plate reaction forces). The methodology is not clearly described (which is a huge problem itself), but it appears that rest of the parameters (e.g., joint stiffness, material properties, muscle forces, etc.) were estimated during simulation to fit the data (plate force, I assume). Thus, it seems that all the important anatomical parameters were changed to fit the fingertip force (plate force) data – which means that the estimated muscle force values are not really “validated”. Since the model has numerous parameters that are “free” to change, the estimated muscle forces and other parameters (e.g., joint stiffness, tendon stiffness, etc.) are just one of possible combination of parameter values that result in the measured fingertip forces.

Minor comments

1. Joint stiffness – stiffness values used in this study were selected arbitrary, and are not based on literature. First of all, why the unit is N/m? Joint stiffness should be defined as N m/rad. Authors said this is a stiffness of springs at the joints – then how are these springs connected (e.g., moment arm - which would critically affect its function)?

Please refer to numerous previous studies on finger joint stiffness (e.g., Milner and Franklin, 1998; Kamper et al., 2002; Jindrich et al., 2004).

2. Page 11: “Validation of validity” – please revise. “Validation of model performance”?

3. MTJ measurement: Note that this represents a kinematic property, which in principle cannot measure any kinetic aspects. For instance, even if MTJ displacement of ED was zero (in Actions 2 – 5), it does NOT mean that ED was not activated. Previous studies show that the extensor muscles are always activated, albeit to a lower degree, during flexion movements.

4. Again, why no intrinsic muscles were considered at all? Intrinsic hand muscles were found to play an important role in force production.

5. Page 12 - Fig. 7: change “flexure” in the caption to “flexion”. Also it is not clear what data these graphs display – in the text (line 221 – 222), it is mentioned that “rigid plate reaction forces of the model were equal to the experimental fingertip forces (Fig 7)”. However, only one line is shown in each figure – what is shown here then? Does it mean that the experimental data and simulation results are in perfect match (which is highly unlikely)?

6. Page 14 - Fig. 9: This figure is not very informative – although the color bar shows a range of different colors, the picture only shows (or appears to show) one color – dark blue.

Also why did the authors show stress distribution throughout the tendons? It would be much more important to show stress at the FINGER PULLEYS (where most stress ruptures happen) – first of all, were the pulleys modeled at all? If so, how was it modeled?

7. The reviewer thinks that the authors may want to focus on utilizing the information obtained from ultrasound imaging (MTJ excursion) and see how that is correlated to the changes in the measured fingertip forces – FE model used in this study is too complex and contains too many unknown parameters to result in any meaningful estimation of muscle forces.

Reviewer #2: Review of PLOS One manuscript PONE-D-21-23164, “Analysis on synergistic cocontraction of extrinsic finger flexors and extensors during Flexion movements: a Finite Element Digital Human Hand Model”

General Comments:

The manuscript describes a model of the human hand performed using limited in vivo experimentation and finite element modeling. The study details are lacking. The results are may be interesting with potential implication. However, the reader is left to question some important details that are missing from manuscript and which are essential for determining if the model is indeed valid (as stated in manuscript) and how that model is any different from already existing human hand models.

Specific comments:

Abstract

Occasionally/many acronyms are defined only in abstract but should also be defined in body of manuscript, e.g. MTJ

Introduction

Line 83: The reader is confused, as the stated purpose is illogical and does not make sense in that the purpose of the paper can not be to develop anything. Rather the purpose of the paper is “to describe the development of an FE-DHHM….” Or the purpose of this paper is to “describe a study to develop an FE-DHHM…”

“The purpose of this paper is to develop a FE-DHHM including phalanges, metacarpals, solid unit tendons and ligaments by combining finite element techniques with ultrasound imaging to measure the MTJ displacements techniques to achieve accurate loading of the MTJ displacements of different muscles under the same movement.”

Lines 89-90: “….and also has a wide range of applications in clinical and rehabilitation fields”

Line 111: what is meant by “cells”? Do authors mean “element”, as this is a finite element model ?

Line 113: what were the “material parameters and boundary conditions defined” ? This specific information would be useful to readers to understand how model was implemented.

Lines 192-193 and 212-213: What exactly is meant by “validation of validity” ? Does not validity stem from validation and so without validation there is no validity ? So, again, specifically, what is meant here by “validation of validity”? How is validation of validity part of methods section but also part of results ? Reader is confused.

Lines 214-215: The table title is “Table 2 MTJ displacements of each muscle during flexion movements (mm)” But then immediately after table 2, sentence reads “Table 2 demonstrates the fingertip forces and MTJ displacements of each muscle…” But all the values in Table 2 have units of millimeters (as stated in title), so how are any of the values in Table 2 forces ? Reader is confused.

Lines 264-266: “FDS and FD contracted together to provide power; while the ED had a non-zero muscle force and acted as an antagonist.” How were forces apportioned between FDS, FD and ED ? This mathematics to determine apportionment is not described in methods section, except for lines 202 and 203 where says “after several fits and adjustments”, and so reader is left to wonder how the values in this Figure 11 were determined ? Was there any physics or mechanics behind the fits and adjustments? Or simply curve matching ?

Line 282-284 seems to say that material parameters are a result of the model. Yet, Table 1 indicates the material parameters were taken from references 26 and 20. So what is difference between table 1 and table 3, and how can material parameters be both input to the model and output from the model ?

Line 239 and Figure 8: this figure suggest the force at finger tip varies with time over 200 milliseconds of the simulation/experiment. What was the sample rate of the force measurement device shown in figure 3 so reader can understand if the variation in finger tip force is meaningful ?

Regarding results, specifically figures 9-12, reader is having difficult time understanding if any of the results are valid as no comparisons are made to in vivo data or even experimental/literature data.

The manuscript suggests some lack of rationality in the model material parameters (line 378). The manuscript suggests some utility of the model (lines 89-90 and 390-391). However, the reader is left to question utility of a model with lack of rationality and without explicitly stated or apparent utility.

6. PLOS authors have the option to publish the peer review history of their article (what does this mean?). If published, this will include your full peer review and any attached files.

Reviewer #1: No

Reviewer #2: No

---

## [Author Response · Author response to Decision Letter 0]

11 Jan 2022

Dear Editors and Reviewers:

Thank you for your letter and for the reviewers’ comments concerning our manuscript entitled “PONE-D-21-23164”. Those comments are all valuable and very helpful for revising and improving our paper, as well as the important guiding significance to our researches. We have studied comments carefully and have made correction which we hope meet with approval. Revised portion were marked in red in the paper. The main corrections in the paper and the responds to the editors and reviewers’ comments were addressed point by point below.

Responds to the editors’ comments:

1. Response to comment: Please ensure that your manuscript meets PLOS ONE's style requirements, including those for file naming. 

Response: The manuscript has been modified according to the PLOS ONE's style requirements.

Response: This study is a supported by the National Natural Science Foundation of China(No.11372208, No.31870934). The authors received no specific funding for this work.

“The support from the National Natural Science Foundation of China (No.11372208, No.31870934, No.11972243) was acknowledged.”

Response: Funding information and other funding-related text have been removed from the manuscript. Funding Statement has been updated as “This study is a supported by the National Natural Science Foundation of China(No.11372208, No.31870934). The authors received no specific funding for this work” within cover letter.

Response: The minimal data set is submitted as Supporting Information files.

5. Thank you for your submission to PLOS ONE. Before we can proceed, we kindly ask you to address the following concerns:

We understand that the author served as the volunteer in the study; however, we ask you to present this information in the Methods section of the manuscript. Please revise your Methods section to state all information about ethics committee approval in this section (including the name of the ethics committee), state that the volunteer was the author of the paper, and state any exclusion/inclusion criteria and any relevant demographic information (sex, age, etc.). We appreciate your attention to these queries and look forward to your response.

Response: Relevant information has been indicated in the manuscript:

To ensure uniformity of model and MTJ displacement data, all CT scans and ultrasound experiments were performed by the author as the volunteer. The volunteer was healthy a 30-year-old male with no hand disease or associated neurological disorders. All experimental protocols and methods were performed in accordance with relevant guidelines and regulations, and were approved by the biological and medical ethics committee of Taiyuan University of technology (page 4, lines 91-96).

Responds to the reviewers’ comments:

Reviewer #1: 

Major comments 

1. Response to comment: Authors adopted an interesting approach, looking at the excursion of the musculotendon junction using ultrasound imaging. This could provide important information regarding the action of different muscles during movements. Unfortunately, there are numerous problems with the modeling approach, and the model validation was performed properly, which makes it very difficult to test the validity of the proposed model. For instance, intrinsic hand muscles are not considered (which are critical in force production), and many model parameters are not properly determined. Some important anatomical features, such as the extensor mechanism or finger pulleys, are not even included in the model.

Response: As Reviewer said, building a perfect model of the human hand is very complex and difficult. In our research work, the model of the hand has indeed been heavily and reasonably simplified. For example, the model does not take into account the intrinsic hand muscles and extensor mechanisms due to the overly complex structure of the hand. The intrinsic hand muscles are characterized by small size, short tendon length, and large numbers, which make modeling efforts difficult to be accurate and tendon displacements difficult to measure. Chang et al.(Chang J , Freivalds A , Sharkey N A , et al. Investigation of index finger triggering force using a cadaver experiment: Effects of trigger grip span, contact location, and internal tendon force[J]. Applied Ergonomics, 2017, 65:183-190.) also addressed only FDS and FDP, not intrinsic hand muscles and ED, in their study of tendon force and index finger triggering force using cadavers. Other studies on hand muscle forces using cadavers have faced the same problem (Schuind et al. ,1992; Valero-Cuevas et al., 1998; Schweizer and Hudek ,2011). This is because the intrinsic muscles of cadavers are difficult to be loaded, just like the intrinsic muscles of models in our work. However this simplification still makes the research work relevant in revealing the contraction mechanisms of the hand muscles. However, this simplification still makes these research efforts indispensable in revealing the contractile mechanisms of the hand muscles.

The extensor mechanism is a tendon network that connects the intrinsic muscles to the ED. In existing models of the human hand, the extensor mechanism is defined as a network of elastic lines or bands and is specific to one finger only. The ED tendon model established in this work is three-dimensional and the minimum cell size of MIMICS software is 1 mm. therefore, it is difficult to establish a three-dimensional extensor network for the whole hand under such conditions.

The main function of the finger pulley mechanisms is to keep the tendons close to the bones. The pulleys in existing models of the human hand are either defined as parametric equations or are ignored.The function of the pulleys in the model of this study is realized by the "joint capsules". The expression in the article is incorrect and has been corrected to "finger pulleys"(page 5, line 105).

The focus of this study is on the method of estimating muscle force using musculotendon junction excursion. The accuracy of the model does have a significant impact on the accuracy of the results. More accurate modeling of the anatomical structures will be our further work.

2. Response to comment: More importantly, the model validation seems to be fundamentally flawed. The only measurements made in this study were musculotendon junction (MTJ) movements and fingertip forces (plate reaction forces). The methodology is not clearly described (which is a huge problem itself), but it appears that rest of the parameters (e.g., joint stiffness, material properties, muscle forces, etc.) were estimated during simulation to fit the data (plate force, I assume). Thus, it seems that all the important anatomical parameters were changed to fit the fingertip force (plate force) data – which means that the estimated muscle force values are not really “validated”. Since the model has numerous parameters that are “free” to change, the estimated muscle forces and other parameters (e.g., joint stiffness, tendon stiffness, etc.) are just one of possible combination of parameter values that result in the measured fingertip forces.

Response: The description of the methodology for parameter determination and model validation in this paper is indeed unclear. It has been revised in the text (page6, lines 126-135). A brief description is as follows.

The parameters of the model, except for the elastic modulus of the bones, need to be determined, including the elastic modulus of the tendons (FDS, FDP, ED, etc.), the elastic modulus of the ligaments (finger joint rotation, extensor retinaculum, etc.), and the joint spring elements stiffness.The elastic modulus of the tendons determines the relationship between MTJ displacement and fingertip force. The elastic modulus of the ligaments determines their effectiveness in restraining the tendons and indirectly affects the relationship between MTJ displacement and fingertip force. There are three joint spring elements at each joint, whose function is to maintain joint stability and to provide joint stiffness (this section is described in detail in the first section of the following minor comments). The simplifying assumption for the spring elements is that each spring element at the IP joints has the same stiffness, and each spring element at the MCP joint has the same stiffness. This gives a total of four unknown quantities: the elastic modulus of the tendons, the elastic modulus of the ligaments, the spring unit stiffness at the IP joints, and the spring unit stiffness at the MCP joints. The experimentally measured known quantities are the MTJ displacement and the fingertip force (or flexion pattern), for a total of five sets. Four of these experimental data sets (MTJ displacement-fingertip force) were used to determine the four unknown quantities and one set of data (MTJ displacement-flexion pattern) was used to validate the model after determining the parameters. Thus the parameters of the model were able to be determined.The muscle force is not involved in the above process; it is the result of the calculation after both parameter determination and model validation have been completed.

Minor comments 

1. Response to comment: Joint stiffness – stiffness values used in this study were selected arbitrary, and are not based on literature. First of all, why the unit is N/m? Joint stiffness should be defined as N m/rad. Authors said this is a stiffness of springs at the joints – then how are these springs connected (e.g., moment arm - which would critically affect its function)?

Please refer to numerous previous studies on finger joint stiffness (e.g., Milner and Franklin, 1998; Kamper et al., 2002; Jindrich et al., 2004).

Response: The spring unit connecting the joints in this paper is a different concept from the joint stiffness in the related literatures (Milner and Franklin, 1998; Kamper et al., 2002; Jindrich et al., 2004).

Specifically, the joint in this paper is connected by three spring elements simulating the left and right collateral ligaments and dorsal ligaments at the joint. The three spring elements were all one-dimensional linear elastic elements arranged along the lateral and dorsal midline of the phalanges (page 7, Fig 2A) . And its unit of stiffness is N/m. These three spring units serve two purposes: (1) stabilize the joint. The joint surfaces are held together during joint rotation without misalignment. (2) The combination of the axial forces of the three springs constitutes the stiffness of the joint during rotation. Therefore, this paper defines the spring element stiffness, not the joint stiffness.

The joint stiffness in the relevant literatures (Milner and Franklin, 1998, etc.) is a spring damper model defined as a combination of two-dimensional spring units at the joints. It measures the joint stiffness in Nm/rad by the net joint torque, the angular displacement of the joint and the angular velocity. This model can describe the joint stiffness directly, but it cannot stabilize the joint. Most of the joints in the literatures are defined by hinge constraints.

2. Response to comment: Page 11: “Validation of validity” – please revise. “Validation of model performance”?

Response: "Validation of validity" has been corrected to "model validation" (page 10, line 209).

3. Response to comment: MTJ measurement: Note that this represents a kinematic property, which in principle cannot measure any kinetic aspects. For instance, even if MTJ displacement of ED was zero (in Actions 2-5), it does NOT mean that ED was not activated. Previous studies show that the extensor muscles are always activated, albeit to a lower degree, during flexion movements.

Response: Indeed, the article elaborates that the displacements of the MTJ are divided into two parts: one part is the displacement of the tendon due to the change in position between the bones, and the other part is the tensile deformation that occurs when the tendon transmits force. Thus the MTJ displacement of the ED is approximately zero and the calculated muscle force of the ED is not zero. The co-activation of the ED during flexion movements is described in the Discussion section (lines 343-345).

4. Response to comment: 4. Again, why no intrinsic muscles were considered at all? Intrinsic hand muscles were found to play an important role in force production.

Response: As described in “response to minor comments 1”, intrinsic muscles have the characteristics of small size, short tendons, and large number, which make their modeling difficult to be accurate and tendon displacement difficult to measure. Moreover, previous cadaveric experiments have shown that studies on extrinsic muscles remain indispensable in revealing the contractile mechanisms of hand muscles without considering intrinsic muscles.

5. Response to comment: Page 12 - Fig. 7: change “flexure” in the caption to “flexion”. Also it is not clear what data these graphs display – in the text (line 221 – 222), it is mentioned that “rigid plate reaction forces of the model were equal to the experimental fingertip forces (Fig 7)”. However, only one line is shown in each figure – what is shown here then? Does it mean that the experimental data and simulation results are in perfect match (which is highly unlikely)?

Response: The purpose of Figure 7 is to show the variation of the rigid plate reaction force with time during the model simulation of Action 2-4, where the most important data is the final value of the reaction force. The representation in Figure 7 was inappropriate and has been removed and replaced with table 3 (page 12, line 244).

6. Response to comment: Page 14 - Fig. 9: This figure is not very informative – although the color bar shows a range of different colors, the picture only shows (or appears to show) one color – dark blue.

Also why did the authors show stress distribution throughout the tendons? It would be much more important to show stress at the FINGER PULLEYS (where most stress ruptures happen) – first of all, were the pulleys modeled at all? If so, how was it modeled?

Response: Indeed. Figure 9 shows the stress distribution of the whole tendon. Due to the size of the image, the color change (stress distribution) cannot be clearly shown. The peak stresses in the tendon are present at the joint area, while the stresses in the palm and wrist areas of the tendon are evenly distributed. Figure 9 has been modified to show the stress distribution of the tendon in the finger area (page 15, line 263, Fig 2A).

The pulley at the joint has been modeled as consisting of a ring ligament around the end of the phalanx at the joint. The right and left sides of the annular ligament are bound to the end of the phalanx. Holes are left on the palmar and dorsal sides of the annular ligament for the extensor and flexor muscles to pass through. An error was made in the article and has been corrected (page 7, Fig 2A).

7. Response to comment: The reviewer thinks that the authors may want to focus on utilizing the information obtained from ultrasound imaging (MTJ excursion) and see how that is correlated to the changes in the measured fingertip forces – FE model used in this study is too complex and contains too many unknown parameters to result in any meaningful estimation of muscle forces.

Response: The description of the methodology for parameter determination in this paper is indeed unclear. It has been revised in the text (page 6, lines 126-135). 

The parameters of the model, except for the elastic modulus of the bones, need to be determined, including the elastic modulus of the tendons (FDS, FDP, ED, etc.), the elastic modulus of the ligaments (finger joint rotation, extensor retinaculum, etc.), and the joint spring elements stiffness. The spring elements are simplified as follows: each spring element at the IP joints has the same stiffness, and each spring element at the MCP joint has the same stiffness. This gives a total of four unknown quantities: the elastic modulus of the tendons, the elastic modulus of the ligaments, the spring unit stiffness at the IP joints, and the spring unit stiffness at the MCP joints. The experimentally measured known quantities are the MTJ displacement and the fingertip force (or flexion pattern), for a total of five sets. Four of these experimental data sets (MTJ displacement-fingertip force) were used to determine the four unknown quantities and one set of data (MTJ displacement-flexion pattern) was used to validate the model after determining the parameters. Thus the parameters of the model were able to be determined.The muscle force is not involved in the above process; it is the result of the calculation after both parameter determination and model validation have been completed.

Reviewer #2: Review of PLOS One manuscript PONE-D-21-23164, “Analysis on synergistic cocontraction of extrinsic finger flexors and extensors during Flexion movements: a Finite Element Digital Human Hand Model”

General Comments:

1. Response to comment: The manuscript describes a model of the human hand performed using limited in vivo experimentation and finite element modeling. The study details are lacking. The results are may be interesting with potential implication. However, the reader is left to question some important details that are missing from manuscript and which are essential for determining if the model is indeed valid (as stated in manuscript) and how that model is any different from already existing human hand models.

Response: The article is not clear about the details of the model description, and the following modifications have been made: (1) detailed description of the definition of interaction, material parameters, boundary conditions and loading conditions (pages 6-7, lines 116-152), especially the setting and simplification of the spring element at the joint; (2) detailed description of the method and process of parameter determination and model validation (pages 10-11, lines 209-232).

Specific comments:

1. Response to comment: Abstract

Occasionally/many acronyms are defined only in abstract but should also be defined in body of manuscript, e.g. MTJ

 Response: Definitions of acronyms have been added to the text，such as FE-DHHM (page 3, line 51) and MTJ (page 4, line 84).

Introduction 

2. Response to comment: Line 83: The reader is confused, as the stated purpose is illogical and does not make sense in that the purpose of the paper can not be to develop anything. Rather the purpose of the paper is “to describe the development of an FE-DHHM….” Or the purpose of this paper is to “describe a study to develop an FE-DHHM…”

 “The purpose of this paper is to develop a FE-DHHM including phalanges, metacarpals, solid unit tendons and ligaments by combining finite element techniques with ultrasound imaging to measure the MTJ displacements techniques to achieve accurate loading of the MTJ displacements of different muscles under the same movement.”

 Response: It has been modified to: In this paper, we have achieved accurate loading of Muscle-Tendon Junction (MTJ) displacements of different muscles under the same movement by combining FE-DHHM established by finite element technique and the MTJ displacements measured by ultrasound imaging, which was used to study the synergistic contraction of Flexor Digitorum Superficialis muscle (FDS), Flexor Digitorum Profundus muscle (FDP) and Extensor Digitorum muscle (ED) in flexion movements (page 4, lines 84-89).

3. Response to comment: Lines 89-90: “….and also has a wide range of applications in clinical and rehabilitation fields”

Response: Removed. The model developed in this paper will be used in the next step of work to study burnt deformities of the hand. This is addressed in the Discussion section, but not in detail. So this sentence is deleted.

4. Response to comment: Line 111: what is meant by “cells”? Do authors mean “element”, as this is a finite element model ?

Response: It does mean "element". Corrected (page 6, lines 112).

5. Response to comment: Line 113: what were the “material parameters and boundary conditions defined” ? This specific information would be useful to readers to understand how model was implemented.

Response: The following modifications were made to the above issues. 

(1) Material parameters (page 6, lines 126-135)

The article covers a total of five material parameters, including the elastic modulus of bones, the elastic modulus of tendons, the elastic modulus of ligaments, the stiffness of spring elements at IP joints, and the stiffness of spring elements at MCP joints. Among them, the material parameters of the bones are known. The elastic modulus of tendons and ligaments need to be determined and have been given preset values (table 1). The stiffness of the spring elements at the IP and MCP joints also need to be determined. In total, four parameters need to be determined. 

(2) Boundary conditions and loading conditions (page 6, lines 136-148)

The boundary and loading conditions were divided into flexion without resistance conditions and flexion with resistance conditions according to the flexion experiments. The flexion without resistance conditions were based on fixing the metacarpals, carpal bones, ulna and radius of FE-DHHM and loading displacement loads on the rigid reference points of FDS, FDP and ED. The flexion with resistance conditions were based on the flexion without resistance conditions with the proximal phalanges fixed and a fully fixed rigid plate added at the tip of the FE-DHHM to provide resistance (Fig 2). 

6. Response to comment: Lines 192-193 and 212-213: What exactly is meant by “validation of validity” ? Does not validity stem from validation and so without validation there is no validity ? So, again, specifically, what is meant here by “validation of validity”? How is validation of validity part of methods section but also part of results ? Reader is confused.

Response: "Validation of validity" has been revised to "model validation". The parameters of the model, except for the elastic modulus of the bones, need to be determined, including the elastic modulus of the tendons (FDS, FDP, ED, etc.), the elastic modulus of the ligaments (finger joint rotation, extensor retinaculum, etc.), and the joint spring elements stiffness. The spring elements are simplified as follows: each spring element at the IP joints has the same stiffness, and each spring element at the MCP joint has the same stiffness. This gives a total of four unknown quantities: the elastic modulus of the tendons, the elastic modulus of the ligaments, the spring unit stiffness at the IP joints, and the spring unit stiffness at the MCP joints. The experimentally measured known quantities are the MTJ displacement and the fingertip force (or flexion pattern), for a total of five sets. Four of these experimental data sets (MTJ displacement-fingertip force) were used to determine the four unknown quantities and one set of data (MTJ displacement-flexion pattern) was used to validate the model after determining the parameters. Thus the parameters of the model were able to be determined.The muscle force is not involved in the above process; it is the result of the calculation after both parameter determination and model validation have been completed.

7. Response to comment: Lines 214-215: The table title is “Table 2 MTJ displacements of each muscle during flexion movements (mm)” But then immediately after table 2, sentence reads “Table 2 demonstrates the fingertip forces and MTJ displacements of each muscle…” But all the values in Table 2 have units of millimeters (as stated in title), so how are any of the values in Table 2 forces ? Reader is confused.

Response: The article was misrepresented. Fingertip forces were not present in Table 2. Correction has been made. The fingertip forces are listed in the revised Table 3 (page 12, line 244).

8. Response to comment: Lines 264-266: “FDS and FD contracted together to provide power; while the ED had a non-zero muscle force and acted as an antagonist.” How were forces apportioned between FDS, FD and ED ? This mathematics to determine apportionment is not described in methods section, except for lines 202 and 203 where says “after several fits and adjustments”, and so reader is left to wonder how the values in this Figure 11 were determined ? Was there any physics or mechanics behind the fits and adjustments? Or simply curve matching ?

Respons: The percentage of FDS muscle forces was the largest and gradually increased with increasing fingertip forces (54%-65%), while the percentage of FDP (45%-34%) and ED (0.7%-0.35%) muscle forces gradually decreased (page 17, lines 285-287).

The parameters were determined by comparing the fingertip forces with the rigid plate reaction forces calculated in the model under loads of MTJ displacements. For example, when the rigid plate reaction force was greater than the fingertip force, reducing the elastic modulus of tendon or ligament, or increasing the stiffness of the joint spring unit, the rigid plate reaction force was reduced. The four material parameters have different effects on the reaction force of the rigid plate: the elastic modulus of the tendon and the elastic modulus of the ligament are positively correlated with the rigid plate reaction force, and the effect of the tendon is more significant. The spring element stiffnesses of the IP and MCP joints were negatively correlated with the rigid plate reaction force, and the effect of the spring element was more significant for the MCP joint. Each adjustment of material parameters required fitting four sets of experimental data (MTJ displacement - fingertip force) simultaneously. After repeated calculations, the parameters that meet the accuracy requirements were obtained. 

Figure 11 of the original article showed the variation of muscle force with fingertip force. Muscle force was not involved in the process of parameter determination and model validation, but was the result after all these processes had been completed.

9. Response to comment: Line 282-284 seems to say that material parameters are a result of the model. Yet, Table 1 indicates the material parameters were taken from references 26 and 20. So what is difference between table 1 and table 3, and how can material parameters be both input to the model and output from the model ?

Response: The model was finally built after the material parameters were determined and the model validation was completed. This process was calculated and adjusted several times. The material parameters were actually the result of the "last" calculation of the model. 

Table 1 shows the elastic modulus of tendons and ligaments taken from the preset values of Refs. 26 and 20. In reality, the models of tendons and ligaments have larger dimensions than the real anatomy, for example the tendon model is thicker than the real tendon. This is a problem caused by the accuracy of the model. Therefore the determined values of the elastic modulus of tendons and ligaments in the model (Table 3) are smaller than the preset values (Table 1). And the preset values taken from Refs. 26 and 20 are the baseline for parameter adjustment.

10. Response to comment: Line 239 and Figure 8: this figure suggest the force at finger tip varies with time over 200 milliseconds of the simulation/experiment. What was the sample rate of the force measurement device shown in figure 3 so reader can understand if the variation in finger tip force is meaningful ?

Response: In the experiments, the fingertip forces displayed by the force measurement device were recorded only for the final values (5, 10, 15 and 20 N). In these processes, the change of fingertip forces with time were not concerned. At the same time, the displacements of the MTJs were recorded only for the corresponding final values, which were loaded uniformly as displacement loads in the model by Step time. Figure 8 was focused on showing the final values of the fingertip forces calculated by the model under MTJ displacement loading. Since Figure 8 was prone to ambiguity and misinterpretation, it has been removed and replaced with Table 3 (page 12, line 244).

11. Response to comment: Regarding results, specifically figures 9-12, reader is having difficult time understanding if any of the results are valid as no comparisons are made to in vivo data or even experimental/literature data.

Response: The data underlying the results shown in Figures 9-12 are the "muscle forces". Figure 9 shows the stress distribution of the tendon under muscle force, Figure 10 shows the relationship between maximum stress and muscle force, Figure 11 shows the relationship between muscle force and fingertip force (experimental value), and Figure 12 shows the variation of muscle force over time calculated by the model.

The results regarding the calculation of muscle force in this paper were described in the Discussion section. Firstly, the calculated results of muscle forces in this paper were compared with the in vivo experimental data of tendon forces in the literatures 34 and 35, and the results were similar with a maximum error of no more than 60%.

Secondly, the factors that affect the accuracy of muscle force calculation results were mainly the accuracy of the model and the ultrasound measurement of MTJ displacements. The simplification of the model in terms of both anatomical structure and material parameters affects the accuracy of the calculation results. However, this is unavoidable for digital methods, and the calculated results of the model are still in good agreement with the in vivo experimental data. This indicates that the model developed in this paper is reasonable and the calculation results are valuable.

12. Response to comment: The manuscript suggests some lack of rationality in the model material parameters (line 378). The manuscript suggests some utility of the model (lines 89-90 and 390-391). However, the reader is left to question utility of a model with lack of rationality and without explicitly stated or apparent utility.

Response: Admittedly, there are various models for material properties of soft tissues in the finite element software ABAQUS, such as viscoelastic model, hyperelastic model, and linear elastic model. Although the nonlinear material parameters (viscoelastic model, hyperelastic model, etc.) are closer to the real situation than the linear elastic model, they are complicated to calculate and difficult to amend and determine the parameters. In the research work on finite element methods for soft tissues, the simplification of linear elasticity is also very common and reasonable. This simplification not only makes the calculation easier to converge and reduces the computational time, but is also very beneficial for the amendment and determination of material parameters.

The practicality of the model was not described in detail in the manuscript. In subsequent work, skin and scar models have been developed and combined with FE-DHHM for the study of the formation and treatment of hand deformities after burn injury. Due to space limitations, only a brief description of the utility of FE-DHHM was provided in this manuscript.

---

## [Decision Letter · Decision Letter 1]

28 Feb 2022

PONE-D-21-23164R1Analysis on synergistic cocontraction of extrinsic finger flexors and extensors during Flexion movements: a Finite Element Digital Human Hand ModelPLOS ONE

Dear Dr. An,

Thank you for submitting your manuscript to PLOS ONE. After careful consideration, we feel that it has merit but does not fully meet PLOS ONE’s publication criteria as it currently stands. Therefore, we invite you to submit a revised version of the manuscript that addresses the points raised during the review process. Reviewer 1 raised additional deeper questions regarding the model accuracy and validation.  I understand these are fundamental issues that could be very challenging to address without restarting a whole new modeling endeavor.  One possible resolution may lie in an attempt to investigate and preferably quantify the effects of assumptions made.

We look forward to receiving your revised manuscript.

Kind regards,

Xudong Zhang

Academic Editor

PLOS ONE

Journal Requirements:

Reviewers' comments:

Reviewer's Responses to Questions

**Comments to the Author**

1. If the authors have adequately addressed your comments raised in a previous round of review and you feel that this manuscript is now acceptable for publication, you may indicate that here to bypass the “Comments to the Author” section, enter your conflict of interest statement in the “Confidential to Editor” section, and submit your "Accept" recommendation.

Reviewer #1: (No Response)

Reviewer #2: All comments have been addressed

2. Is the manuscript technically sound, and do the data support the conclusions?

Reviewer #1: Partly

Reviewer #2: Yes

3. Has the statistical analysis been performed appropriately and rigorously? 

Reviewer #1: N/A

Reviewer #2: Yes

4. Have the authors made all data underlying the findings in their manuscript fully available?

Reviewer #1: No

Reviewer #2: Yes

5. Is the manuscript presented in an intelligible fashion and written in standard English?

Reviewer #1: Yes

Reviewer #2: Yes

6. Review Comments to the Author

Reviewer #1: In this study, the authors created a finite element model of the musculoskeletal structure of the human fingers to estimate muscle forces during finger flexion. Authors provided answers to the questions previously raised by the reviewer, but these answers actually led to more significant questions, as listed below:

1. Model inaccuracy: missing muscles and anatomical structures.

Authors responded that the intrinsic hand muscles and extensor mechanism are ‘ignored’ in this model because they are basically ‘difficult to model’. They mentioned one study (Chang et al., 2017) from the applied ergonomics field to provide rationale for excluding intrinsic muscles. However, first, this study was looking at a specific “triggering” motion (concurrent flexion of the DIP and PIP joints without much MCP flexion), which does not require intrinsic action (this is close to intrinsic minus motion). Second, some studies they mentioned (e.g., Valero-Cuevas et al., 1998) did include intrinsic actions, so their answer to this question is basically inaccurate.

More importantly, in my opinion, the authors should focus on providing a proper rationale, i.e., why these muscles or mechanisms were not included in this study (e.g., if their force contribution is negligible) to respond to this reviewer's question. But instead, throughout their response the authors emphasized basically how “difficult” it is to include the intrinsic muscles or the extensor mechanism – the response should be, to properly justify their model selection, why the effects of the intrinsic muscles and/or the extensor mechanism on the model output is small enough (negligible) to exclude these structures. Note that, unfortunately, that will not be the case (i.e., the contribution of intrinsic muscles are quite significant indeed) – see Maier and Hepp-Reymond (1995) and/or Milner and Dhaliwal (2002), which emphasized the importance of intrinsic muscles during force production tasks (similar to what was done in this study).

2. Flawed model validation

The authors now provided details of the model parameter estimation process – which is basically estimating material properties of the tendons, ligaments, and joint stiffness. This is a problematic in itself since all these components are connected in series, which means that the effects of these parameters on measured data (fingertip force/MTJ motion) are intertwined (and cannot be told from each other). In other words, given the motion tested (concurrent flexion of all finger joints), there is no way for the model to distinguish the effects of joint stiffness from those of tendon stiffness. Thus, the outcome may appear to make sense, but there is no way to validate the results (or whether any of these estimated parameter values is reliable). Hypothetically, the solutions obtained by the authors may estimate the movement of MTJ correctly, but could lead to 5-fold overestimation of tendon stiffness in combination with 5-fold underestimation of joint stiffness. Therefore, it is possible that this model will work on the dataset collected in this experiment ('interpolation'), but won't be applicable to any other cases ('extrapolation').

Reviewer #2: All concerns are addressed

All concerns are addressed

All concerns are addressed

All concerns are addressed

All concerns are addressed

7. PLOS authors have the option to publish the peer review history of their article (what does this mean?). If published, this will include your full peer review and any attached files.

Reviewer #1: No

Reviewer #2: No

---

## [Author Response · Author response to Decision Letter 1]

11 Mar 2022

Dear Editors and Reviewers:

Thank you for your letter and for the reviewers’ comments concerning our manuscript entitled “PONE-D-21-23164”. Those comments are all valuable and very helpful for revising and improving our paper, as well as the important guiding significance to our researches. We have studied comments carefully and have made correction which we hope meet with approval. Revised portion were marked in red in the paper. The main corrections in the paper and the responds to the editors and reviewers’ comments were addressed point by point below.

Responds to the reviewers’ comments:

Reviewer #1: 

1. Response to comment: Model inaccuracy: missing muscles and anatomical structures.

Authors responded that the intrinsic hand muscles and extensor mechanism are ‘ignored’ in this model because they are basically ‘difficult to model’. They mentioned one study (Chang et al., 2017) from the applied ergonomics field to provide rationale for excluding intrinsic muscles. However, first, this study was looking at a specific “triggering” motion (concurrent flexion of the DIP and PIP joints without much MCP flexion), which does not require intrinsic action (this is close to intrinsic minus motion). Second, some studies they mentioned (e.g., Valero-Cuevas et al., 1998) did include intrinsic actions, so their answer to this question is basically inaccurate.

More importantly, in my opinion, the authors should focus on providing a proper rationale, i.e., why these muscles or mechanisms were not included in this study (e.g., if their force contribution is negligible) to respond to this reviewer's question. But instead, throughout their response the authors emphasized basically how “difficult” it is to include the intrinsic muscles or the extensor mechanism – the response should be, to properly justify their model selection, why the effects of the intrinsic muscles and/or the extensor mechanism on the model output is small enough (negligible) to exclude these structures. Note that, unfortunately, that will not be the case (i.e., the contribution of intrinsic muscles are quite significant indeed) – see Maier and Hepp-Reymond (1995) and/or Milner and Dhaliwal (2002), which emphasized the importance of intrinsic muscles during force production tasks (similar to what was done in this study).

Response:

Thank you for your comments on our article. The effect of the intrinsic muscles and/or the extensor mechanism on the model of this article was indeed insignificant. The literatures suggest that the contribution of the intrinsic muscles of the hand is relevant to the target task. For example, Li et al. (Li Z M , Zatsiorsky V M , Latash M L . Contribution of the extrinsic and intrinsic hand muscles to the moments in finger joints[J]. Clinical Biomechanics, 2000, 15( 3):203-211.) analyzed the contribution of intrinsic and extrinsic muscles to the knuckle moment using the experimental apparatus shown in Figure 1 in conjunction with a biomechanical model of the index finger and concluded: when the point of force application was on the distal phalanx, the force of the INT of the index finger accounted for only 22.5% of the combined FDP and FDS force. The moment contribution of the INT at the MCP joint of the index finger was only 12.4%. In further work, Li et al. (Li Z M , Zatsiorsky V M , Latash M L . The effect of finger extensor mechanism on the flexor force during isometric tasks[J]. Journal of Biomechanics, 2001, 34(8):1097-1102.) stated: when the point of force application was at the distal phalanx, the extrinsic flexor muscles flexor digitorum profundus (FDP) and flexor digitorum superficialis (FDS) accounted for over 80% of the summed force of all flexors, and therefore were the major contributors to the joint flexion at the distal interphalangeal (DIP), proximal interphalangeal (PIP), and metacarpophalangeal (MCP) joints. When the point of force application was at the DIP joint, the FDS accounted for more than 70% of the total force of all flexors, and was the major contributor to the PIP and MCP joint flexion. When the force of application was at the PIP joint, the intrinsic muscle group was the major contributor for MCP flexion, accounting for more than 70% of the combined force of all flexors. The results suggested that the effects of the extensor mechanism on the flexors were relatively small when the location of force application was distal to the PIP joint. When the external force was applied proximally to the PIP joint, the extensor mechanism had large influence on force production of all flexors. 

Maier and Hepp-Reymond (M.A. Maier, M.C.Hepp-Reymond. EMG activation patterns during force production in precision grip[J]. Experimental Brain Research, 1995,103:108-122.) reported that the intrinsic muscles of the index finger and thumb were closely related to the low isometric forces generated between the thumb and index finger in the precision grasp experiment (Figure 2).

It can be seen that the contribution of the intrinsic muscles of the hand is related to the target task. The target task (fingertip force) addressed in this paper is very similar to the distal phalanx loading in the work of Li et al. Under this task, the extrinsic muscles are the main contributors to joint flexion of the DIP, PIP and MCP joints (accounting for more than 80% of the total force of all flexors). Therefore, the simplification of the intrinsic muscles and the extensor mechanism in the modeling of this paper is justified.

Admittedly, the intrinsic muscles are important components of the hand muscles, and a whole-hand model containing the intrinsic muscles has still not been created. The modeling and loading of intrinsic muscles in the whole hand model is worthy of consideration and is something we intend to carry out in further work (page 18, lines 308-315).

2 .Response to comment: Flawed model validation

The authors now provided details of the model parameter estimation process – which is basically estimating material properties of the tendons, ligaments, and joint stiffness. This is a problematic in itself since all these components are connected in series, which means that the effects of these parameters on measured data (fingertip force/MTJ motion) are intertwined (and cannot be told from each other). In other words, given the motion tested (concurrent flexion of all finger joints), there is no way for the model to distinguish the effects of joint stiffness from those of tendon stiffness. Thus, the outcome may appear to make sense, but there is no way to validate the results (or whether any of these estimated parameter values is reliable). Hypothetically, the solutions obtained by the authors may estimate the movement of MTJ correctly, but could lead to 5-fold overestimation of tendon stiffness in combination with 5-fold underestimation of joint stiffness. Therefore, it is possible that this model will work on the dataset collected in this experiment ('interpolation'), but won't be applicable to any other cases ('extrapolation').

Response: 

The influence of the material parameters to be determined for the model on the measured data can be divided into two cases: the stiffness of tendons and ligaments was positively correlated (to different degrees) with the fingertip force, and the stiffness of the two groups of joints (IP and MCP joints) was negatively correlated (to different degrees) with the fingertip force. All parameters were approximated iteratively using experimental data on the basis of initial values (literature data). And four sets of data were sufficient to determine the four unknowns. The effect of each parameter on the measured data was not "proportional", so that the the solutions for parameter estimation would not lead to "5-fold overestimation of tendon stiffness in combination with 5-fold underestimation of joint stiffness ". Otherwise the determined material parameters would not satisfy the " MTJ-flexion pattern" data in the model validation, even if the four sets of "MTJ-fingertip force" data were satisfied simultaneously. In fact, the model validation process could be regarded as the "extrapolation" of the model after the parameters were determined.

---

## [Decision Letter · Decision Letter 2]

25 Apr 2022

Analysis on synergistic cocontraction of extrinsic finger flexors and extensors during Flexion movements: a Finite Element Digital Human Hand Model

PONE-D-21-23164R2

Dear Dr. An,

We’re pleased to inform you that your manuscript has been judged scientifically suitable for publication and will be formally accepted for publication once it meets all outstanding technical requirements.

Kind regards,

Xudong Zhang

Academic Editor

PLOS ONE

Additional Editor Comments (optional):

Reviewers' comments:

Reviewer's Responses to Questions

**Comments to the Author**

1. If the authors have adequately addressed your comments raised in a previous round of review and you feel that this manuscript is now acceptable for publication, you may indicate that here to bypass the “Comments to the Author” section, enter your conflict of interest statement in the “Confidential to Editor” section, and submit your "Accept" recommendation.

Reviewer #1: (No Response)

Reviewer #2: (No Response)

2. Is the manuscript technically sound, and do the data support the conclusions?

Reviewer #1: Partly

Reviewer #2: Yes

3. Has the statistical analysis been performed appropriately and rigorously? 

Reviewer #1: (No Response)

Reviewer #2: Yes

4. Have the authors made all data underlying the findings in their manuscript fully available?

Reviewer #1: No

Reviewer #2: Yes

5. Is the manuscript presented in an intelligible fashion and written in standard English?

Reviewer #1: Yes

Reviewer #2: Yes

6. Review Comments to the Author

Reviewer #1: No comment.

Reviewer #2: I have no further questions/comments for authors beyond the comments and questions raised by the other reviewers. When I read the response, I am not quite sure the other reviewer's comments were adequately addressed in the response.

7. PLOS authors have the option to publish the peer review history of their article (what does this mean?). If published, this will include your full peer review and any attached files.

Reviewer #1: No

Reviewer #2: No

---

## [Editor Report · Acceptance letter]

2 May 2022

PONE-D-21-23164R2 

Analysis on synergistic cocontraction of extrinsic finger flexors and extensors during Flexion movements: a Finite Element Digital Human Hand Model 

Dear Dr. An:

I'm pleased to inform you that your manuscript has been deemed suitable for publication in PLOS ONE. Congratulations! Your manuscript is now with our production department. 

Kind regards, 

on behalf of

Dr. Xudong Zhang 

Academic Editor

PLOS ONE